# Neural Shuffle-Exchange Networks − Sequence Processing in O($n$ log $n$) Time

**Kārlis Freivalds, Emīls Ozoliņš, Agris Šostaks**
Institute of Mathematics and Computer Science
University of Latvia
Raina bulvaris 29, Riga, LV-1459, Latvia
{Karlis.Freivalds, Emils.Ozolins, Agris.Sostaks}@lumii.lv

## Abstract

A key requirement in sequence to sequence processing is the modeling of long range dependencies. To this end, a vast majority of the state-of-the-art models use attention mechanism which is of O($n^2$) complexity that leads to slow execution for long sequences.

We introduce a new Shuffle-Exchange neural network model for sequence to sequence tasks which have O(log $n$) depth and O($n$ log $n$) total complexity. We show that this model is powerful enough to infer efficient algorithms for common algorithmic benchmarks including sorting, addition and multiplication. We evaluate our architecture on the challenging LAMBADA question answering dataset and compare it with the state-of-the-art models which use attention. Our model achieves competitive accuracy and scales to sequences with more than a hundred thousand of elements.

We are confident that the proposed model has the potential for building more efficient architectures for processing large interrelated data in language modeling, music generation and other application domains.

## 1 Introduction

A key requirement in sequence to sequence processing is the modeling of long range dependencies. Such dependencies occur in natural language when the meaning of some word depends on other words in the same or some previous sentence. There are important cases, e.g., to resolve coreferences, when such distant information may not be disregarded. A similar phenomenon occurs in music, where a common motif may reappear throughout the entire piece and it should be kept coherent by any applied transformation (Huang et al., 2019). Dealing with long range dependencies require processing very long sequences (several pages of text or the entire musical composition) in a manner that aggregates information from their distant parts.

Aggregation of distant information is even more important for algorithmic tasks where each output symbol typically depends on every input symbol. The goal for algorithm synthesis is to derive an algorithm from given input-output examples which are often given as sequences. Algorithmic tasks are especially challenging due to the need for processing sequences of unlimited length. Also, generalization plays an important role since training is often performed on short sequences but testing on long ones.

Currently the best neural network architectures do not scale well with the sequence length. A large fraction of them uses the attention mechanism which has quadratic complexity depending on the sequence length. These models can be easily trained on length 512 or so but become very slow and memory hungry on longer sequences.

On sequential algorithmic tasks, the best architecture in the respect of a variety of learnable tasks and generalization to longer sequences is the improved (Freivalds and Liepins, 2018) Neural GPU (Kaiser and Sutskever, 2015). It has O($n$) convolutional layers where each layer performs O($n$) operations where $n$ is the input length. This architecture can represent algorithms of running time $\Theta(n^2)$ but to learn faster algorithms, for example of complexity $O(n \log n)$, a fundamentally new approach is needed.

In this paper, we propose a new differentiable architecture for sequence processing tasks that has depth O(log $n$) and allows modeling of any dependencies in the sequence. This architecture is derived from the Shuffle-Exchange network used for packet routing in the field of computer networks (Dally and Towles, 2004).

We empirically validate our model on algorithmic tasks and the LAMBADA question answering task(Paperno et al., 2016). Our model is able to synthesize $O(n \log n)$ time algorithms for common benchmark tasks such as copy, reversal, sorting and long binary addition which generalize to longer sequences. On the LAMBADA task, our model scores second best in terms of accuracy, losing only to the Universal Transformer(Dehghani et al., 2018), but is significantly faster and able to process 32x longer sequences than the Universal Transformer.

## 2 Related Work

Common tools for sequence processing tasks are recurrent networks, in particular, LSTM (Hochreiter and Schmidhuber, 1997) and GRU networks (Cho et al., 2014). They can efficiently process sequences of any length and have the ability to remember arbitrary long dependencies. But they process symbols one by one and have limited state memory, hence can remember only a limited number of such dependencies. They are successful at natural language processing but too weak for nontrivial algorithmic tasks. Grid LSTM (Kalchbrenner et al., 2015) allows creating a multilayered recurrent structure that is more powerful, able to learn more complex tasks such as addition and memorization at the expense of increased running time.

Convolutional architectures can be used for sequence processing tasks (Gehring et al., 2017). But convolutions are inherently local − the value of a particular neuron depends on a small neighborhood of the previous layer, so it is common to augment the network with the attention mechanism. Attention allows combining information from any two locations of the sequence in a single step. Attention mechanism has become a standard choice in numerous neural models including Transformer (Vaswani et al., 2017) and BERT (Devlin et al., 2018) which achieve state-of-the-art accuracy in NLP and related tasks. Although efficient for moderately long sequences, the complexity of the attention mechanism is quadratic depending on the input length and does not scale to longer sequences.

Researchers have recognized the need for processing long sequences and are searching for ways to overcome the complexity of attention. An obvious way is to cut the sequence into short segments and use attention only within the segment boundaries (Al-Rfou et al., 2018). To, at least partially, recover the lost information, recurrent connections can be added between the segments (Dai et al., 2019). Child et al. (2019) reduce the complexity of attention to O($n\sqrt{n}$) by attending only to a small predetermined subset of locations. Star-Transformer (Guo et al., 2019) sparsifies attention even more by pushing all the long range dependency information through one central node and reaches linear time performance. Clark and Gardner (2017) enable document level question answering by preselecting the paragraph most likely to contain the answer.

A different way to capture long range structure is to increase the receptive field of convolution by using dilated (atrous) convolution, where the convolution mask is spread out at regular spatial intervals. Dilated architectures have achieved great success in image segmentation (Yu and Koltun, 2015) and audio generation (van den Oord et al., 2016) but are hard to apply to algorithmic tasks that require generalization to longer sequences. That would require layers with shared weights but different dilation patterns, a setting that is not yet explored. Also, it is important to avoid congestion (for example, that may arise in tree-like structures) when a lot of long range information is forced to travel through a few nodes. Both these problems are elegantly solved with the proposed architecture.

The design of memory access is crucial for learning algorithmic tasks, see (Kant, 2018) for a good overview. To access memory, attention mechanism over the input sequence is used in Pointer Networks (Vinyals et al., 2015). Specialized memory modules, which are controlled by a mechanism

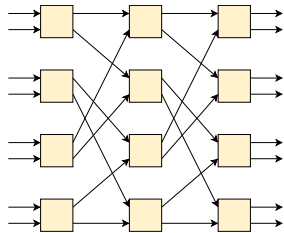

Figure 1: Shuffle-Exchange network.

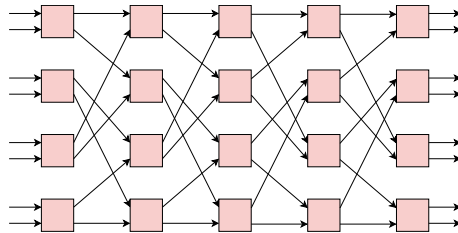

Figure 2: Beneš network.

similar to attention, are used by Neural Turing Machine (Graves et al., 2014) and Differentiable Neural Computer (Graves et al., 2016).

Neural GPU (Kaiser and Sutskever, 2015) utilizes active memory (Kaiser and Bengio, 2016) where computation is coupled with memory access. This architecture is simple and fast and can learn fairly complicated algorithms such as long number addition and multiplication. The computation and memory coupling introduces a limitation: for the information to travel from one end of the sequence to the other, o($n$) layers are required which result in $\Omega(n^2)$ total complexity. The flow of information is facilitated by introducing diagonal gates in (Freivalds and Liepins, 2018) that improves training and generalization, but does not address the performance problem caused by many layers.

The goal of inferring algorithms with running time O($n \log n$) is pursued in (Nowak et al., 2018) which use the divide and conquer paradigm with learnable split and merge steps. A hierarchical memory layout with logarithmic access time is introduced in Andrychowicz and Kurach (2016), however, the model is discrete and has to be trained with reinforcement learning to achieve the claimed performance. Neural programmer-interpreters (Reed and de Freitas, 2016) can learn very efficient algorithms but often require program traces during training. Neural Random-Access Machines (Kurach et al., 2016) introduce a memory addressing scheme potentially allowing constant time access facilitated by discretization. Discrete models are hard to train, but our model, on the contrary, is continuous and differentiable and yet allows synthesizing O($n \log n$) time algorithms.

## 3 Shuffle-Exchange Networks

Routing messages from many sources to many destinations is a well-explored topic in the area of computer networks where several kinds of sparse architectures have been developed to connect two sets of devices. The Shuffle-Exchange network[1] has a regular layered structure and serves best as a prototype for a neural network. Shuffle-Exchange network consists of repeated application of two stages - shuffle and exchange. Fig. 1 shows a Shuffle-Exchange network for routing 8 messages. Messages arrive on the left side, flow through the network layers and arrive at the rightmost nodes.

We consider Shuffle-Exchange networks with $2^k$ inputs and pad the input data to the nearest power of two. First comes the exchange stage where elements are divided into adjacent pairs and each pair is passed through a switch. The switch contains logic to select which input is routed to which output. The shuffle stage follows(depicted as arrows in the figures), where messages are permuted according to the perfect-shuffle permutation. The perfect-shuffle is often employed to shuffle a deck of cards by splitting the deck into two halves and then interleaving the halves. In this permutation, the destination address is a cyclic bit shift(left or right) of the source address. The network for routing $2^k$ messages contain $k$ exchange stages and $k-1$ shuffle stages. It is proven that switches can always be programmed in a way to connect any source to any destination through the network (Dally and Towles, 2004).

But the throughput of the Shuffle-Exchange network is limited − it may not be possible to route several messages simultaneously. A better design for multiple message routing is the Beneš network (see Fig. 2). The Beneš network is formed by connecting a Shuffle-Exchange network with its mirror copy[2]. The mirror copy is obtained by reversing the direction of bit shift in the destination address

calculation. Beneš network has $2k-1$ exchange stages and $2k-2$ shuffle stages. Such a network can route $2^k$ messages in any input-to-output permutation (Dally and Towles, 2004).

## 4 The Model

We propose a neural network analogue of the Beneš network where we replace each switch with a learnable 2-to-2 function. The input to the neural network is a sequence of cells of length $2^k$ where each cell is a vector of size $m$. The network consists of alternating Switch and Shuffle layers. The Switch Layer corresponds to a column of boxes in Fig. 1 and Fig. 2 and the Shuffle Layer corresponds to the links between the box columns.

In the Switch Layer, we divide the cells into adjacent non-overlapping pairs and apply Switch Unit to each pair[3]. The Switch Unit is similar to Gated Recurrent Unit(GRU) but it has two inputs $[s^1, s^2]$ and two outputs $[s_o^1, s_o^2]$. It contains two reset gates, one for each output. The reset gate performs the computing logic of the unit($\sigma$ and tanh nonlinearities just keep the values in range) and it is important that each output uses a separate reset gate for the unit to produce unrelated outputs. Technically, creating the pairs is implemented as reshaping the sequence $s$ into a twice shorter sequence where each new cell concatenates two adjacent cells $[s^1, s^2]$ along the feature dimension. The Switch Unit is defined as follows:

$$s = [s^1, s^2]$$
$$r^1 = \sigma(W_r^1 s + B_r^1)$$
$$r^2 = \sigma(W_r^2 s + B_r^2)$$
$$c^1 = \tanh(W_c^1(r^1 \odot s) + B_c^1)$$
$$c^2 = \tanh(W_c^2(r^2 \odot s) + B_c^2)$$
$$u = \sigma(W_u s + B_u)$$
$$\tilde{s} = \text{swapHalf}(s^1, s^2)$$
$$[s_o^1, s_o^2] = u \odot \tilde{s} + (1-u) \odot [c^1, c^2]$$

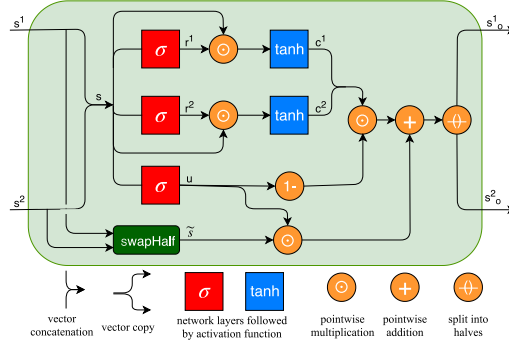

In the above equations, $W_r$ and $W_u$ are weight matrices of size $2m \times 2m$, $W_c$ are weight matrices of size $2m \times m$, $B$ are bias vectors; these are the parameters that will be learned; $\odot$ denotes element-wise vector multiplication and $\sigma$ is the sigmoid function.

The function swapHalf splits the values of $s_1$ and $s_2$ into two halves along the feature dimension and swaps their second halves:

$$\text{swapHalf}\left(\begin{bmatrix} a \\ b \end{bmatrix}, \begin{bmatrix} c \\ d \end{bmatrix}\right) = \begin{bmatrix} \begin{bmatrix} a \\ d \end{bmatrix}, \begin{bmatrix} c \\ b \end{bmatrix} \end{bmatrix}$$

The motivation for swapHalf is to encourage the unit to perform one of its two default actions — return the two inputs unchanged or to swap them. The update gate $u$, which is borrowed from GRU, is responsible for this. For GRU, there is only one default action and its update gate performs a straight-through copy. To facilitate both actions of the Switch Unit, we perform the straight-through copy in the first half of the feature maps and swapped copy in the second half. Such a fixed assignment to maps works better than introducing another gate for action selection, see ablation study below. A similar fixed assignment was found beneficial in (Freivalds and Liepins, 2018) for introducing diagonal update gates.

The Shuffle Layer permutes cells according to bit rotation permutation, namely $s[x] = s[\text{rotate}(x, k)]$, where $\text{rotate}(x, k)$ performs cyclic bit shift of $x$ by one position, where $x$ is treated as a binary number of length $k$. Left rotation is used in the first part of the Beneš network, right in the second (the difference is insignificant if we apply the rotations the other way). Shuffle Layer has no learnable parameters. The whole network is organized by connecting these two kinds of layers in the pattern of the Beneš network. A deeper architecture can be obtained by stacking several blocks where each is

in the form of the Beneš network. In such a case, the last switch layer of every but the final Beneš block is omitted. We typically use two stacked Beneš blocks in our models, except for the simple algorithmic tasks which use one block. Please see Fig. 3 of the whole model containing two Beneš blocks on input length 16, $k = 4$. We employ residual skip connections between blocks where a scaled input of each Switch Layer is added to the input of a corresponding layer in the next Beneš block. The scaling factor is a learnable parameter resembling the forget gate of LSTM.

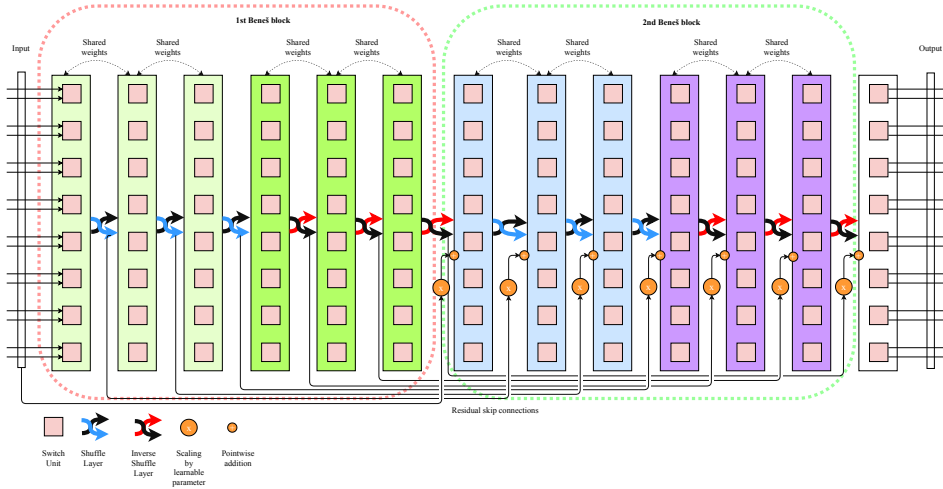

Figure 3: Neural Shuffle-Exchange network with 2 Beneš blocks.

We use shared weights for every consecutive $k - 1$ layers(shown with colors in Fig. 3). The last layer of the last Beneš block has non-shared weights. Weight sharing is required to obtain generalization to longer sequences but we use it always, also for tasks not requiring generalization. This way we reduce the number of learnable parameters without an observable decline of accuracy. See more analysis in the Appendix.

## 5   Evaluation

In this section, we evaluate the proposed architecture on algorithmic tasks and the LAMBADA question answering task (Paperno et al., 2016). We have implemented the proposed architecture in TensorFlow. The code is available at `https://github.com/LUMII-Syslab/shuffle-exchange`. The neural network model for evaluation consists of an embedding layer where each symbol of the input is mapped to a vector of length $m$, one or more Beneš blocks and the output layer which performs a linear transformation to the required number of classes with a softmax cross-entropy loss for each symbol independently(except for the LAMBADA task which will be described later). All models are trained on a single Nvidia RTX 2080 Ti (11GB) GPU with Adam optimizer (Kingma and Ba, 2014).

### 5.1   Algorithmic tasks

Let us evaluate the Shuffle-Exchange network to see if it can infer O($n \log n$) time algorithms purely from input-output examples. We consider sequence duplication, reversal, long binary addition, long binary multiplication, sorting. These tasks are common benchmarks in several papers including (Kalchbrenner et al., 2015; Zaremba and Sutskever, 2015; Zaremba et al., 2016; Joulin and Mikolov, 2015; Grefenstette et al., 2015; Kaiser and Sutskever, 2015; Freivalds and Liepins, 2018; Dehghani et al., 2018). There are known O($n \log n$) time algorithms for these tasks (Brent and Kung, 1982; Ajtai et al., 1983; Seiferas, 2009; Harvey and Van Der Hoeven, 2019), although for sorting and multiplication all known O($n \log n$) time algorithms have huge constants hidden by the O notation and are not realistically usable in practice.

The proposed architecture performs very well on simple tasks(all except multiplication) − it achieves **100% test accuracy** on examples of length it was trained on, so the main focus is to explore how

well these tasks generalize to longer inputs. Multiplication is a considerably harder task than the rest; from the existing architectures only the Neural GPU (Kaiser and Sutskever, 2015; Freivalds and Liepins, 2018) is able to learn it purely from input-output examples. The Shuffle-Exchange architecture can learn multiplication up to certain input length (depending on the model size) but the solution does not generalize to longer examples. Since it is the hardest task, we analyze it separately and use it for tuning our model and ablation study.

We use dataset generators and curriculum learning from (Freivalds and Liepins, 2018). For training we instantiate several models for different sequence lengths (powers of 2) sharing the same weights and train each example on the smallest instance it fits.

Let us consider the simple tasks first. We train them on inputs of length 64 and test on 8x longer instances. A small model comprising one Beneš block and 192 feature maps suffice for these tasks. For the duplication, reversal, sorting we use a fixed alphabet of symbols in range 1-12. Fig. 4 shows the generalization of simple tasks to sequences of length 512 vs. training step. The graphs show the accuracy on the test set averaged over 5 training runs. We see that the duplication and reversal tasks converge in a few hundred steps and generalize perfectly, the addition task quickly reaches 90% accuracy(we measure the fraction of correctly predicted output symbols) and then converges to 98% in about 10k steps. The sorting task reaches about 95% generalization accuracy.

It is interesting that this architecture (of depth $2 \log n$) is able to learn a sorting algorithm that generalizes. Although the best known sorting circuits have depth $O(\log n)$ (Ajtai et al., 1983; Seiferas, 2009), they are huge, with estimated depth over $100 \log n$. Simple sorting circuits have depth $\Theta(\log^2 n)$ (Knuth, 1973). Apparently, the inferred algorithm takes advantage of the discrete nature and limited range of the alphabet whereas the mentioned sorting circuits work in a comparison based model that is applicable to any numeric values. Indeed, when we try increasing the alphabet size, training becomes slower and a larger model is required at some point.

The multiplication task can be trained up to quite a large length (longer examples require a larger model and longer training), the solution generalizes to other examples of the same length but not to longer examples. We chose to demonstrate the multiplication task on sequences of length 128 for which a model can be trained in a reasonably short time. A model with 384 feature maps reaches 100% accuracy on the test in about 50K steps, see Fig. 10.

The generalization to longer sequences of the Shuffle-Exchange network is shown in Fig. 6. We compare it to the optimized implementation of the Neural GPU with diagonal gates (DNGPU) by Freivalds and Liepins (2018) which is currently the best in this respect. Shuffle-Exchange network provides a similar generalization on duplication and reversal tasks but performs better on the addition and sorting tasks. Both models were trained for 40k steps on length up to 64 symbols. A possible explanation of why the multiplication task does not generalize is that there may not exist a simple $O(n \log n)$ time algorithm for it. The only currently known asymptotically optimal algorithm (Harvey and Van Der Hoeven, 2019) is galactic but practical algorithms, for example based on the Fast Fourier Transform, have complexity $O(n \log n \log \log n)$ that is slightly more than available within our model. We have tried training a model of depth $O(\log^2 n)$ but were not able to obtain better generalization. Such model is relatively deep for the sequence lengths on which it can be realistically trained(for example, we get 113 switch layers for sequence length 256) on and it seems unable to recognize the proper scaling of $O(\log^2 n)$ instead of $cn$ for some constant $c$.

A distinct advantage of the Shuffle-Exchange model is its speed. Fig. 5 shows the comparison of the running time of the learned binary addition algorithm (or any other algorithm having the same sized model) with the proposed model and DNGPU. We use 96 feature maps for both models which is enough for both of them to learn binary addition. The time complexity of our model is $O(n \log n)$ vs. $O(n^2)$ for DNGPU which is reflected in the measured running time. Also, DNGPU demands a lot more GPU memory − we were able to evaluate it only up to length 16K, whereas the Shuffle-Exchange model is able to process sequences of two million symbols in about 5 seconds.

## 5.2 LAMBADA question answering

The goal of the LAMBADA task is to predict a given target word from its broad context (on average 4.6 sentences collected from novels). The sentences in the LAMBADA dataset (Paperno et al., 2016) are specially selected such that giving the right answer requires examining the whole passage. In 81% cases of the test set the target word can be found in the text and we follow a common strategy

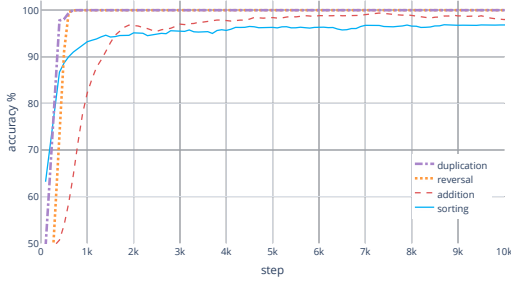
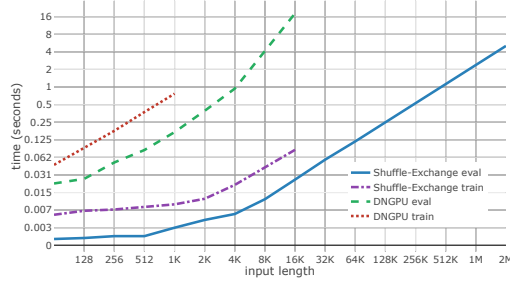

Figure 4: Accuracy on length 512 vs training step.

Figure 5: Evaluation and training time comparison with DNGPU (log scale).

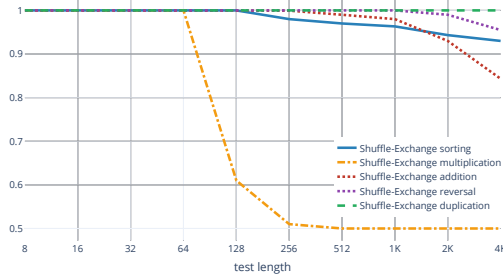

(a) Shuffle-Exchange

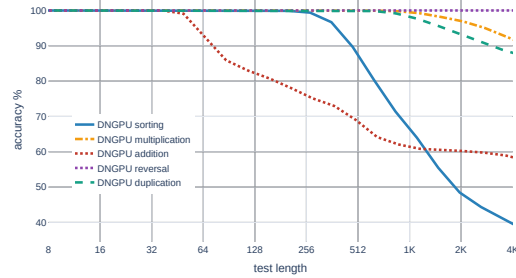

(b) DNGPU

Figure 6: Generalization to longer sequences.

(Chu et al., 2017; Dehghani et al., 2018) to choose the target word as one from the text. Obviously, the answer will be wrong in the remaining cases so the obtained accuracy will not exceed 81%.

We instantiate the model for input length 128 (almost all test examples fit into this length) and pad the input sequence to that length by placing the sequence at a random position and adding zeros on both ends. Without randomization, the model overfits. We use a pretrained fastText 1M English word embedding (Mikolov et al., 2018) for the input words. The embedding layer is followed by 2 Beneš blocks with 384 feature maps. To perform the answer selection as a word from the text, the final layer of the network is constructed differently. Each symbol of the output is linearly mapped to a single scalar and we use softmax loss over the obtained sequence to select the position of the answer word.

In Table 1 we give our results in the context of previous works. Our architecture is able to score better than Gated-Attention Reader (Chu et al., 2017) and loses only to Universal Transformer (Dehghani et al., 2018). Both these networks use attention mechanism enclosed in a highly elaborate neural model, so it is remarkable that our model of much simpler architecture can provide a competitive accuracy.

On the other hand, our model is significantly faster and has better scaling to long sequences. In Fig. 7 we compare the training and execution time of our model to the Universal Transformer. We use the official Universal Transformer implementation from the Tensor2Tensor library (Vaswani et al., 2018) and measure the time for one training or evaluation step on a single sequence. For both models, we use configurations that reach the best test accuracy. We use the base configuration (as mentioned in Dehghani et al. (2018)) for the Universal Transformer which has 152M parameters and the Shuffle-Exchange network with 384 feature maps and 2 Beneš blocks and total parameter count 33M. It's worth mentioning that our model is about 5x smaller than Universal Transformer. We perform the evaluation on sequence lengths that are able to fit in the 11GB of GPU memory. We can see that the Shuffle-Exchange network is significantly faster and can be applied up to 32x longer sequences using the same amount of GPU memory.

It is interesting to note that, in contrast to Transformers, our architecture does not need positional embeddings. It can learn positional embedding itself if required. Assume that the input has a marked position(end-of-line marker, for example) and consider the binary tree of paths from it to the nodes at

depth $\log(n)$. Each leaf of this tree can be uniquely labeled according to left-or-right choices on the path connecting it to the root. Such labeling can be learned by the first $\log(n)$ layers of the network and the rest of the network can use this information as positional embedding.

Table 1: Accuracy on LAMBADA word prediction task

| Model | Test accuracy (%) |
|---|---|
| Random word from passage (Paperno et al., 2016) | 1.6 |
| Gated-Attention Reader (Chu et al., 2017) | 49.0 |
| Shuffle-Exchange network (this work) | **52.28** |
| Universal Transformer (Dehghani et al., 2018) | 56.0 |
| Human performance (Chu et al., 2017) | 86.0 |

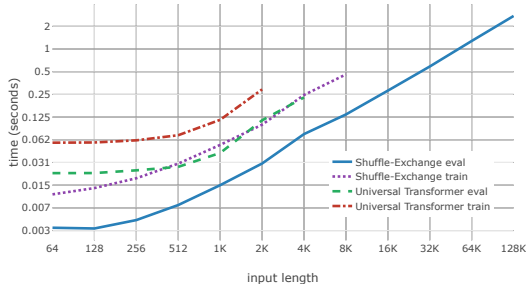

Figure 7: Evaluation and training time of Shuffle-Exchange and Universal Transformer (log scale).

## 5.3 Ablation study

We have studied several modifications of our architecture and confirmed that the presented form is the best. We verified the need for the key features by removing them one by one and comparing with a baseline version on the multiplication, LAMBADA, addition and sorting tasks (Fig. 8). We considered the following ablations: using the identity function instead of swapHalf (without swap), removing residual connections(without residual), disabling the idea by Beneš to use two shuffle directions(without Beneš), using an additional swap gate in place of swapHalf function which chooses a straight-through or a swapped input for the update gate (swap gate). Additionally, we explore two versions where the Switch Unit is replaced with two fully connected layers with ReLU and twice the number of feature maps in the middle(ReLU on both FC layers does not work well). The first ablation replaces the entire unit (Two FC layers), the second one replaces only the part involving reset gates but keeps the update gate(Two FC layers+gate).

We can see that the baseline version gives the best accuracy overall. The differences are most pronounced on the multiplication task, on other tasks all versions, except the one without the gate, perform reasonably well.

We have also evaluated the effect of the network depth(measured in the Beneš block count) and feature map count on the test accuracy, see Figures 9 and 10. There are no surprises − a larger and deeper network is generally better.

## 6  Conclusions

We have introduced a Shuffle-Exchange neural network model that has O($\log n$) depth and O($n \log n$) total complexity and allows modeling any dependencies in the sequence. We have shown that this model can successfully synthesize nontrivial O($n \log n$) time algorithms with good generalization. The Shuffle-Exchange model can serve as an alternative to the attention mechanism with better scaling to long sequences. Although we obtained slightly lower accuracy on the LAMBADA question answering task, our model is much simpler than the winning Universal Transformer and has fewer parameters. We are looking forward to implementing more elaborate architectures combining Shuffle-Exchange blocks with other kinds of layers to achieve unmatched accuracy for long sequence tasks.

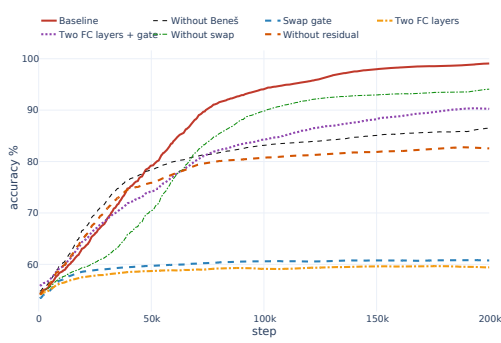

(a) Multiplication task

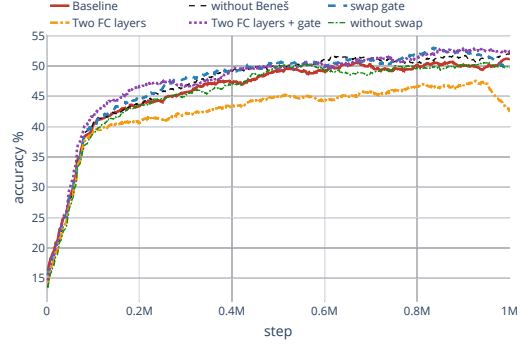

(b) LAMBADA task

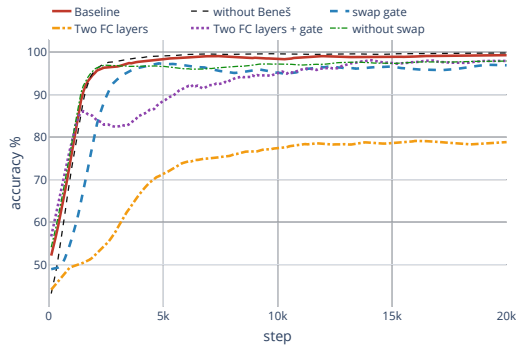

(c) Addition task

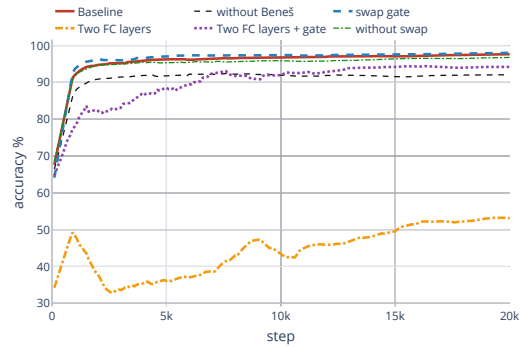

(d) Sorting task

Figure 8: Ablation study.

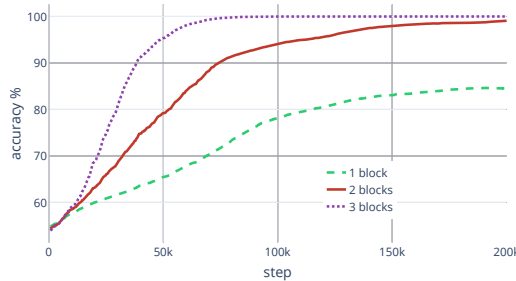

(a) Multiplication task

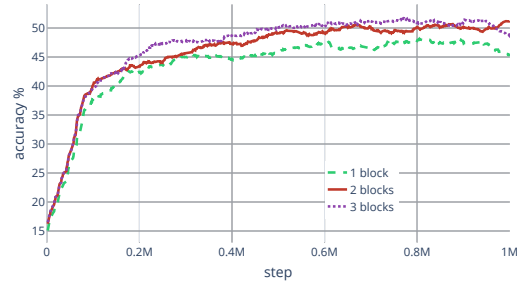

(b) LAMBADA task

Figure 9: Test accuracy depending on the block count.

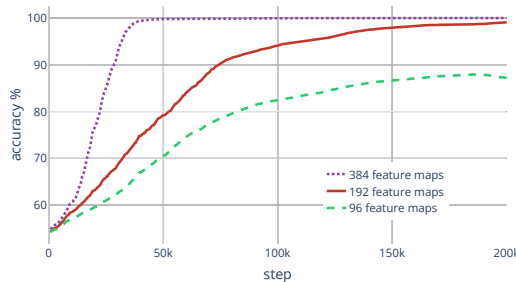

(a) Multiplication task

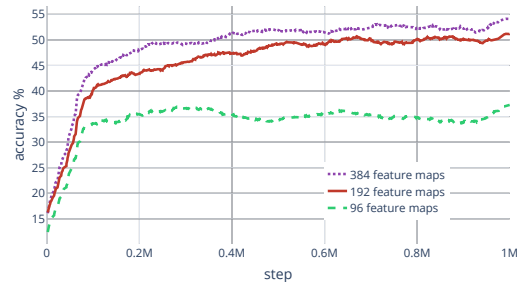

(b) LAMBADA task

Figure 10: Test accuracy depending on the feature map count.

## Acknowledgements

We would like to thank Renārs Liepiņš for the valuable discussion regarding the subject matter of the paper, the IMCS UL Scientific Cloud for the computing power and Leo Trukšāns for the technical support. We sincerely thank all the reviewers for their comments and suggestions. This research is funded by the Latvian Council of Science, project No. lzp-2018/1-0327.

## Footnotes

[1] Also called Omega network.

[2] In the literature, the Butterfly network is typically used instead which is isomorphic to the Shuffle-Exchange network.

[3]Identical transformation is applied to all pairs of the sequence.

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
