[Supplementary Material]

# Supplementary Material
# Neural Shuffle-Exchange Networks − Sequence Processing in O($n$ log $n$) Time

**Kārlis Freivalds, Emīls Ozoliņš, Agris Šostaks**
Institute of Mathematics and Computer Science
University of Latvia
Raina bulvaris 29, Riga, LV-1459, Latvia
{Karlis.Freivalds, Emils.Ozolins, Agris.Sostaks}@lumii.lv

## Alternative Weight Sharing

We have evaluated a different weight sharing scheme, see the drawing below, which shares as little as possible weights but still enables generalization to longer sequences. This version performs similarly to our default choice presented in Fig. 3 but has more parameters.

Minimal weight sharing.

The following picture shows a comparison of different weight sharing schemes. All the different sharings with 192 feature maps give a similar accuracy. But increasing the number of feature maps does increase accuracy. Note that we have included the embedding layer in the parameter count which has 11.5M parameters for 192 feature maps(embedding size 192, 60K vocabulary) and 23M parameters for 384 feature maps(embedding size 384, 60K vocabulary). So the weights of the Shuffle-Exchange network give only a fraction of the total parameters.

Effect of weight sharing on LAMBADA task.