[Reviews · NeurIPS 2019]

Reviewer 1



This paper focuses on the problem of scalably processing long sequences, in particular, processing sequences without requiring O(n^2) space and time complexity. This is an important goal, as many existing architectures consider pairwise interactions (as in the Transformer) or require inputs to propagate one step at a time (as in the Neural GPU), leading to large runtime complexity. To address this issue, the authors take inspiration from computer network routing, using a shuffle-exchange network to route connections between neural network nodes. They then build on top of this idea, adding gates similar to a GRU and other variations, and show that their architecture performs well on a variety of tasks. ### Originality and Significance The idea of using computer network routing designs as a basis for neural network connectivity patterns is, to my knowledge, a novel contribution of this work. Their architecture is also notable for the fact that it does not involve any discrete sampling steps, but is instead entirely continuous and thus can be trained using standard backpropagation. This work is clearly inspired by Neural GPUs, as the authors state in this work. The Neural GPU architecture also avoids discrete choices, instead applying a series of convolutions across the input. However, because of the nature of the convolutions, running a Neural GPU takes a large amount of memory. The architecture presented in this work appears to be a potential replacement for Neural GPUs in situations with long inputs. The authors also show that their architecture can approach the Universal Transformer in performance on a word prediction task, indicating its versatility. The authors do a good job of surveying prior work in sequence processing architectures. One thing missing is a mention of explicit program-synthesis-based architectures such as the Neural Programmer-Interpreter (Reed and Freitas, 2016) (among others), which can learn very efficient algorithms but often require program traces during training. The authors mention dilated-convolution architectures such as WaveNet (Oord et al. 2016), but state that "it is impossible to cover a sequence of a varying length with a fixed dilation pattern" (lines 81-82). I'm not sure what this statement means. It seems that a dilated-convolution pattern could be extended similarly to how a perfect-shuffle pattern can be, as long as weight-sharing is set up properly and edges are handled appropriately (i.e., increase the dilation amount by a factor of two for each layer, and share weights across those layers). In fact, a shuffle-exchange network looks very similar to a dilated convolution when you consider only the edges connected to a single input or output. It may be true that the shuffle-exchange network has better inductive biases because of how states are propagated, but simply asserting that a dilated-convolution-based architecture is impossible seems like too strong of a claim. Another note on related work: the discussion of Divide-and-Conquer Networks (on lines 98-99) may not be quite correct. That paper appears to describe using a neural network to determine how to split up the inputs, so it does not have a fixed algorithm structure (and instead it requires reinforcement learning). ### Quality The model itself presented in the paper seems reasonable, although I am not sure I fully understand the intuition for parts of it. The cell equations are modeled after a GRU, but if the update gate is set to 1 (preventing an update) the first half of each input is kept as-is and the second halves are swapped with each other. The authors note that this fixed assignment works better than an additional gate, but it seems strange to to have one part of the input always be swapped and the other part always not be swapped by default. The choice of experiments seems reasonable, including both a set of algorithmic tasks and a natural-language-processing task. For the majority of the algorithmic tasks, the authors show how performance varies over training for their model, and also how runtime varies with input length compared with other models. However, notably missing is a comparison of how the accuracy of their model compares with the accuracies of the other models (such as the Neural GPU) as a function of training progress, or a comparison of generalization performance between models (i.e. generalization performance of each model as a function of input length). This seems like important information for comparing this architecture to the existing models. The paper also includes an ablation study, comparing multiple versions of their model for the multiplication task in particular. These results are very interesting, but I think this section could be improved by exploring a few additional variants of their architecture. In particular, the authors mention in section 4 that using an additional gate to determine swapping or not performs worse; this seems like a perfect candidate for an ablation study. (The authors do evaluate using an identity without swapping, but this isn't quite the same.) Also, it would be very interesting to see how a simple fully-connected network with a couple of layers would work when used instead of the gated update equations. This would help clarify how much of their model's performance is due to the shuffle-exchange/Benes topology, and how much is due to the specific architecture design. ### Clarity Overall, I felt that the paper was clearly written and well organized. A few things I noticed: - Line 48: typo "loosing" should be "losing". - Line 116: A less technical introduction to the perfect-shuffle might be nice (i.e. stating that it corresponds to splitting the input in half, then interleaving) in addition to the bit-shift explanation. Also, perfect shuffles are well defined for non-powers-of-two, but the provided description only makes sense for powers of two. Perhaps this would be a good place to mention that you are restricting your attention to sequences of length 2^k, and padding inputs to allow this. - Lines 134-135: I'm not sure what the claim "largely responsible for the power of the unit" is based on. Is there evidence for that? - Lines 143-144: The notation used made me think about binomial coefficients; perhaps use square brackets instead of parentheses. - Lines 160-170: This description would be much clearer if accompanied by a picture. In particular, this would help make the weight sharing and residual connections easier to understand; I initially misinterpreted the connectivity of the residual blocks and was confused when glancing at the provided code. - Figure 3: Perhaps it would be easier to interpret this figure if shown on a log scale (and maybe show 1-accuracy instead of accuracy); most of the graph is empty. It would also be useful to see the accuracy for the length-64 examples, for reference. - Line 255: typo "looses" should be "loses" ### Citations Reed, Scott, and Nando De Freitas. "Neural programmer-interpreters." ICLR (2016). Van Den Oord, Aäron, et al. "WaveNet: A generative model for raw audio." SSW 125 (2016). ### EDIT: After reading author response The diagram clarifying the weight sharing is very helpful for understanding the full structure of the model, thank you for providing it. The plots comparing the generalization performance of the Shuffle-Exchange and Neural GPU are very interesting. In particular, I find it remarkable that the two models show qualitatively different generalization curves for the different tasks; Shuffle-Exchange seems not to smoothly generalize for multiplication, but shows remarkable generalization performance for the other tasks, whereas the Neural GPU generalizes better for multiplication but shows a sharp drop-off in generalization performance for the duplication and addition. This suggests that this architecture allows fundamentally different types of behavior in addition to the efficiency gains. I also greatly appreciate the additional ablation results, which show how the specific architecture choices affect behavior. Adding a swap gate seems to improve accuracy for sorting, which makes intuitive sense to me, but doesn't seem to help for the more challenging multiplication task (as the authors noted in the original submission). As most of my original concerns have been addressed, I am increasing my rating from 7 to 8. I look forward to reading the final version of the paper.

Reviewer 2



This paper describes a neural network for sequence processing whose topology is inspired by sparse routing networks. The primary advantage of the proposed architecture is the ability to process length $n$ sequences with $O(n \log n)$ complexity, which is a significant performance improvement over attention based models (Transformers, NeuralGPU, etc), which typically have complexity $O(n^2)$. The authors demonstrate the effectiveness of their model using previously studied synthetic algorithmic tasks (sorting, binary multiplication, etc), and the LAMBADA QA dataset. I really enjoyed the ideas presented in this paper. The connection to message routing networks is quite interesting. To the best of my knowledge this is novel, and, similar to other "neuralifications" of CS concepts [1, 2, 3], could be a productive area of future research. The results of the ablation study are interesting, although it is unfortunate they are not more conclusive given that the most interesting properties (swap, residual, Benes) had little effect on the LAMBADA dataset. It would be very useful to know if similar results are observed on tasks where the model does learn algorithms which generalize to longer sequences (unlike multiplication), and I would appreciate including these results in the main body of the paper. My biggest concern is that the architecture is somewhat difficult to understand. I've never encountered Shuffle-Exchange Networks before, and do not imagine this is a topic that most of the NeurIPS community is familiar with. I believe it would be very beneficial if the authors included additional exposition or an Appendix that gave a more detailed description of the architecture. I highlight specific points of confusion below. I found the description of the weight sharing confusing. Based on my reading, there are essentially two sets of Switch Unit parameters. The first for layers $1, k, 2k-1, \ldots$, and the second for layers $2, \ldots, k-1, k+1, 2k-2, 2k, \ldots$ Is that correct? I find it interesting that the architecture does not require any sort of positional encoding. Perhaps this is due to the fixed nature of the shuffling stage, or perhaps due to some misunderstanding of the architecture on my part. It would be helpful if the authors could comment on this? How exactly are sequences given to the model? In all the diagrams, each switch has two inputs, so I'm assuming that given a sequence of length $2^k$, [x1, x2, x3, x4, ….] then adjacent pairs [(x1, x2), (x3, x4), …] are given to each Switch Unit (of which there are only $2^{k-1}$ with shared parameters per layer). Is this correct? Line 134: "Rest gate is" -> "The reset gate is" Lines 254-258: The wording here shows a bit of bias, and should be adjusted. In particular, the absolute performance differences between the Gated-Attention Reader and Shuffle-Exchange Network (3.28) is smaller than between the Shuffle-Exchange Network and the Universal Transformer (3.72), so it is not appropriate to use "looses only slightly" exclusively when referring to the later. #### Post Author Response I thank the authors for their helpful response, it cleared up several ambiguities for me. I find the architectural diagram included within very helpful and would strongly encourage the author's include it in subsequent versions. I would also encourage the authors to include the additional ablation studies from other algorithm tasks, as well as figures depicting generalization as a function of sequence length compared the the NeuralGPU (possibly in an appendix). #### References 1. Learning to Transduce with Unbounded Memory, http://papers.nips.cc/paper/5648-learning-to-transduce-with-unbounded-memory 2. Meta-Learning Neural Bloom Filters, http://proceedings.mlr.press/v97/rae19a/rae19a.pdf 3. Neural Random Access Machines, https://ai.google/research/pubs/pub45398

Reviewer 3



The proposed architecture is well motivated and interesting, and the paper is well written, however, the evaluation is less convincing. The new architecture is evaluated on a set of arithmetic tasks where it has generalization capability on par or worse than the NeuralGPU (depending on the task) but is significantly faster. Sometimes, we want to make a trade off between generalization and how fast the model is, but it is not clear how the lower generalization performance compares to other similar models and so whether it is still relatively good. The model can generalize to longer sequences better though. The second task the model is applied to is a real and difficult natural language task Lambada where the model performs well compared to other models but does not outperform the Universal Transformer, which is fine. My concern is that this task has short inputs (up to 128 tokens) and so doesn’t really necessitate the main selling point of the new model architecture: scalability to much longer sequences and faster run time / better complexity. Hence, even though the eval time is faster than the Universal Transformer (by about 3x?? at 128 tokens) this doesn’t support that this new architecture will work well in the setting it’s designed for (for long inputs). The paper has a nice ablation study of the network (without Benes, without swap, without residual) that shows differences for the multiplication task but the ablation study on the Lambada task doesn’t seem to support that all the new features of the model are important.

[Author Response · NeurIPS 2019]

Thank you for your valuable comments and suggestions. Please see the following drawing of the complete model(input
length 16, $k = 4$). Weight sharing is shown with colored layers. White layers have unique weights. You are right
that for a sequence of length $2^k$, [x1, x2, x3, x4, ....] adjacent pairs [(x1, x2), (x3, x4), ...] are given to each Switch
Unit. There are $2^{k-1}$ pairs with shared weights per layer. We designed the sharing to be minimal such that the model
generalizes to longer inputs. We found that this schema helps to reduce the parameter count without accuracy loss
also on tasks not requiring generalization, see the Lambada figure. It could be possible to reduce more parameters by
sharing more weights. We will analyze this in the final paper.

Effect of weight sharing on Lambada

Our architecture compared to Diagonal Neural GPU has competitive generalization on longer test sequences. Both
   models were trained for 40k steps on length up to 64 symbols.

The main purpose of the update gate is to stabilize the gradient flow through the layers(similar to LSTM and GRU). It
does not increase the expressive power of the unit much since its logic can be simulated by the other parts of the unit.
See the following ablation experiments. They display multiplication accuracy trained and tested on length 128, addition
and sorting accuracy trained on 64, tested on 512. Three additional ablations are provided:(swap gate) an additional
swap gate is introduced; (Two FC layers) the entire Switch Unit is replaced with two fully connected layers with ReLU
and twice the number of feature maps in the middle;(Two FC layers+gate) the part involving reset gates is replaced with
two FC layers. ReLU on both FC layers does not work well. To maximize the variety of solvable tasks, the hardest ones
should be given more impact in ablation study. We have used about 15 tasks in total for tuning our model.

This architecture can learn positional information by itself. Assume that the input has a marked position(end-of-line
marker, for example) and consider the binary tree of paths from it to nodes at depth log($n$). Each leaf of this tree can be
uniquely labeled according to left-or-right choices on the path connecting it to the root.

We would like to note that learning fast algorithms is a considerably harder task than learning slow algorithms(you
may try challenging your students to come up from scratch with a log-depth circuit to add long binary numbers). Fast
algorithms are often considerably more complex and, remarkably, that our architecture can learn them. Also, it is
natural that the more complex algorithms learned by our model generalize worse than possibly simpler ones learned by
the Neural GPU. Therefore we consider our work a significant contribution in the field of algorithm learning, regardless
that we do not obtain better accuracies.

We agree that showing the benefits of our architecture on a real-wold task with long sequences is an important future
work.

We will include the analysis given here and take care of the other review suggestions in the final paper.

[Meta-Review · NeurIPS 2019]

The paper presents a way to incorporate sparse routing networks into the transformer architecture to reduce the computation cost of attention for long sequences. The reviewers acknowledge that the idea is novel and the experiments suggest that the proposed architecture is potentially useful. However, the experiments do not demonstrate improved efficiency or accuracy on real world tasks with long sequences. Comparison with Transformer architectures that make use of sparse attention is lacking. Hence, I recommend acceptance as a poster.